# Spatiotemporal Analysis of Urban Green Spatial Vitality and the Corresponding Influencing Factors: A Case Study of Chengdu, China

**Qidi Dong** [1,2] **, Jun Cai** [2,*] **, Shuo Chen** [2] **, Pengman He** [2] **and Xuli Chen** [1]

1 School of Art and Design, Xihua University, Chengdu 610039, China
2 College of Landscape Architecture, Sichuan Agricultural University, Chengdu 611130, China
* Correspondence: caijun@sicau.edu.cn

**Abstract:** Green space integrates diversified urban functions, and the analysis of its utilization, can help improve the vitality of the social and economic development of cities while contributing to the important goal of enhancing urban green space (UGS) planning and management. In this study, the heat value obtained from Baidu heat maps was used as an external representation of spatial vitality, and the factors that influence vitality were analyzed from two dimensions, i.e., the inner and outer space characteristics of green space, using spatial big data such as points of interest (POIs), Open Street Map (OSM) and online review data. The findings indicated that green space and urban resources in Chengdu are highly centralized. That is, a high road network density and nearby transportation facilities make it easier for visitors to reach parks, while peripheral functional density also plays a role in promoting vitality; additionally, reasonable and moderate space and functional layouts are conducive to the development of green space. In addition, our study integrates the tour experience index, which has a strong positive impact on vitality, to better reflect the human-oriented characteristics of green space, which are of great relevance to the construction and renewal of human space in UGSs.

**Keywords:** landscape architecture; urban green space; spatial vitality; big data; spatiotemporal analysis; influencing factors of vitality; humanity site

## 1. Introduction

Green space is an important type of public open space in a city and an important place for urban residents to carry out leisure, sports, gatherings and other social activities. These spaces serve people, and only people who are willing to use them can benefit from their value [1]. With rapid urbanization, urban construction provides vast green space for people, but it also produces problems such as a low utilization rate and lack of vitality, leading these green spaces to become unused "green deserts" [2]. The high-frequency use of urban green space (UGS) can not only provide vitality for the social and economic development of the city but also serve as an important target for UGS planning and management [3].

At present, research on spatial vitality is relatively mature, and related research has covered a wide range of urban space scenarios. In her classic works, Jane Jacobs began to reflect on the influence of orthodox urban planning on the utilization rate of urban space and pointed out that changes in spatial scale and the level of quality have a great impact on the frequency of urban public space use [2]. At the same time, Jan Gehl analyzed the influence of activities, slow traffic and outdoor stay time on urban vitality and pointed out that utilization frequency and utilization mode are important factors in generating living space [4]. However, in the literature, the research methods selected by scholars are mainly qualitative, and the data sources rely mainly on surveys [5,6], expert scoring [7,8] and other methods that require many human and material resources. For example, Chinese scholars Wang and Jiang developed an evaluation model of urban public spatial vitality



that integrates the sensory, social, economic and cultural levels and takes into account 60 indicators that fully cover all types of spatial vitality and elements of urban public spaces. However, this method requires considerable time to complete research on a certain space, and there may be only a small amount of data to obtain. Defects with regard to data updates, such as a slow speed and a long research period, make it difficult to extend such research to conduct a more generalized analysis of UGS use [9,10].

Based on these problems, in recent years, big data reflecting the integration of the relationship between people and cities have gradually entered the field of urban research, providing new ideas for spatial frequency research in terms of spatiotemporal breadth, refinement and timeliness, and has been widely applied by scholars [11,12]. In terms of data development, use and methodology, a large number of geospatial positioning data, such as location-based services (LBSs) [13,14], heat maps [15], and mobile signal data [16], have been widely used in research on the use of urban spaces. Scholars have analyzed the characteristics of people's aggregation and differentiation in urban spaces through spatiotemporal and dynamic analyses. Some scholars have used point of interest (POI) [17,18] and social network data [19,20] to connect the aggregation degree of urban functional facilities, positioning information, consumer comments and other information with the intensity of human activities, thus explaining the use of urban space. On this basis, to analyze the influencing factors of spatial vitality, scholars have applied a variety of comprehensive data analysis methods to urban spaces. For example, Li et al. evaluated the supply-and-demand relationship of urban park use by using POI data and social network check-in data and clarified the key factors affecting vitality through regression analysis [21,22]. Qin et al. combined multisource data such as mobile signal data, POI data, land use data and road network data and studied the influencing factors of spatial vitality and its constituent factors through quantitative analysis [23]. According to relevant studies, accessibility is the main factor affecting the use of space; that is, high-density green space can bring more vitality to these areas [24,25]. Kuang et al. built an evaluation model for the use of the heat of urban parks and found that accessibility accounted for the highest proportion of the evaluation content, and access distance factors, such as being within 10 miles, 15 min and 2 km, were important indicators of vitality. Above these distances, vitality gradually decays [26]. Yoon concluded that the site selection and accessibility of UGSs are the main factors affecting the frequency of space use; appropriate location brings high vitality to green spaces, and this vitality feeds back to the surrounding streets and drives regional development [27]. Regarding other aspects, in addition to distance and other external factors, the UGS scale [28], spatial function and activity setting [29], plant configuration [28] and other internal factors influence the vitality of UGSs. Optimizing these factors can provide diverse and vital UGSs. However, previous research on the influencing factors of vitality has focused mainly on green space itself, while quantitative research on the external environment and its relationship is lacking, and green spaces have become isolated.

On this basis, in view of the existing study space which has been based on less research data, the research time was long. This study expects to discover a quantitative, accurate and real-time method to analyze spatial vitality and to use quantitative data identification to understand the spatial vitality as well as the spatial and temporal distribution of UGSs. Subsequently, in view of the current lack of any external environment analysis of the problems associated with research regarding UGSs, the goal of this study is to discover ways of linking the inner and outer spaces of UGSs to strengthen the connection between the inner environment of the UGS and the outer urban space. Finally, this research is expected to reveal the key factors that affect the spatial vitality of UGSs with the aim of serving as a reference for the construction, renewal and improvement of the spatial vitality of UGSs in the future.

## 2. Materials and Methods

### 2.1. Study Area

Chengdu, the capital of Sichuan Province in China, has been known as the "the land of abundance" since ancient times. According to the statistical results of the Chengdu Park Construction Administration, the green coverage area of Chengdu reached 528.71 km$^2$ in 2020, and the per capita green area was 14.9 m$^2$. In this study, we focus on UGSs within the Third Ring Road of Chengdu with a large human flow. This area measures approximately 196 km$^2$ and is the main residential area of Chengdu residents. In this way, the adverse impact of accessibility problems on the vitality of UGSs can be excluded to a certain extent. The density of UGSs in this area is relatively large, and the structure of the green space system mainly features dotted and linear green space, which includes various kinds of small parks and greenbelts located along the river system and roads (Figure 1).

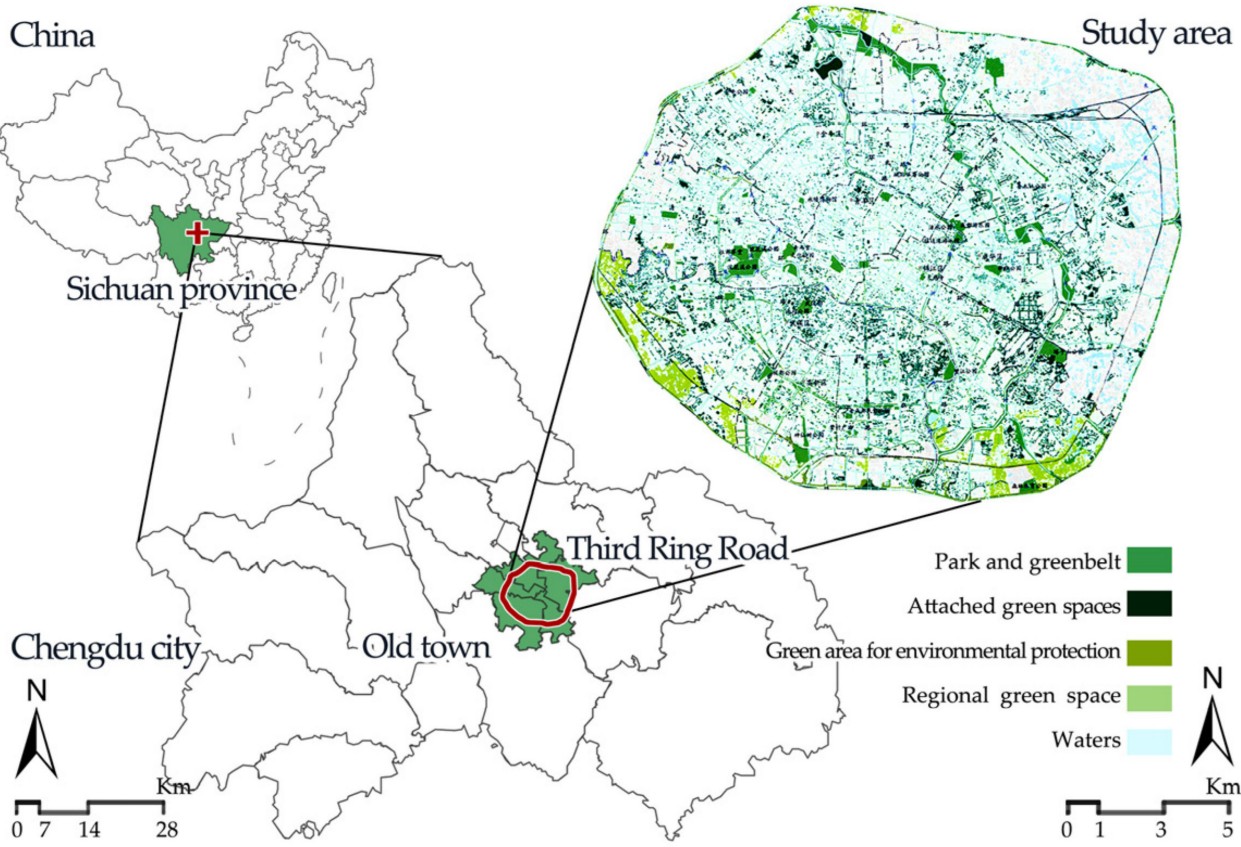

**Figure 1.** Geographical location of the Chengdu Third Ring Road area.

### 2.2. Data Sources

The main data types referenced by this study are as follows. (1) Spatial vitality data were collected from the Baidu heat map. In terms of data time selection, first, the data from 2018 were selected to avoid the impact of COVID-19 on travel. Second, previous studies on urban crowd activity have shown that the degree of crowd aggregation shows periodic changes during the week, and the distribution of urban crowds on weekdays and weekends is significantly different [30–32]. Therefore, spatial data from 7:00 to 24:00 on 8 March (a work day) and 10 March (a rest day) were selected to calculate UGS vitality. (2) POI data derived from Amap included 8 categories and 14 subcategories of data pertaining to topics, such as companies, shopping services, and financial and insurance services and covering information such as names, classifications, addresses, and latitude and longitude measurements; these data were used mainly for the analysis of the external function characteristics of UGS. (3) Open Street Map (OSM) data were collected from the OSM

website. These data were used mainly to calculate external road network density and to identify land patches. (4) Online review data from Ctrip were used. This study collected online review data from Ctrip from July 2022 and before, including information such as user name, time, review text and score, and used them mainly to analyze visitor perceptions. (5) The building information data were obtained from secondhand housing trading websites such as Anjuke, as well as encyclopedia websites; these data were used mainly to calculate the built environment surrounding UGS. (6) Chengdu Park Urban Green Space System Planning (2019–2035) was selected as a basis for green space classification to facilitate the identification and division of green space. The specific method and process of data use are detailed in the analysis (Table 1).

**Table 1.** Data name, date and source.

| Name | Date | Data Source |
| --- | --- | --- |
| Spatial vitality data | 8 and 10 March 2018 | Baidu heat maps (https://lbsyun.baidu.com/ (accessed 8 and 10 March 2018)) |
| POI data | 2021 | Amap (https://lbs.amap.com/ (accessed 27 May 2021)) |
| OSM data | 2022 | Openstreetmap (https://www.openstreetmap.org/(accessed 22 July 2022)) |
| Online review data | All information before 2022 | Ctrip (https://you.ctrip.com/ (accessed 20 July 2022)) |
| Building information data | 2022 | Anjuke (https://www.anjuke.com/ (accessed 5 July 2022)) |
| UGS System Planning | - | Chengdu government (http://www.chengdu.gov.cn/ (accessed 20 July 2022)) |
| Investigations data | 27/29 January 2019 | Investigation |

*2.3. Research Framework*

Based on the data and methods discussed above, we developed the research framework for UGS spatial vitality used in this study (Figure 2). First, the multisource data were organized, georegistered and visualized to complete the data preprocessing with the aim of completing the construction of the evaluation framework for the factors influencing spatial vitality. Subsequently, by processing the average value of the Baidu heat maps and superimposing them on the UGS system planning and OSM data, the average value of spatial vitality within the Third Ring Road of Chengdu was obtained. After combining these data with related indicators, spatio-temporal quantitative analysis was conducted. Finally, using spatial vitality as the dependent variable and 18 indicators as independent variables, multiple linear regression was used to discover the main factors affecting spatial vitality, and suggestions and inspirations for future UGS spatial development are proposed on this basis.

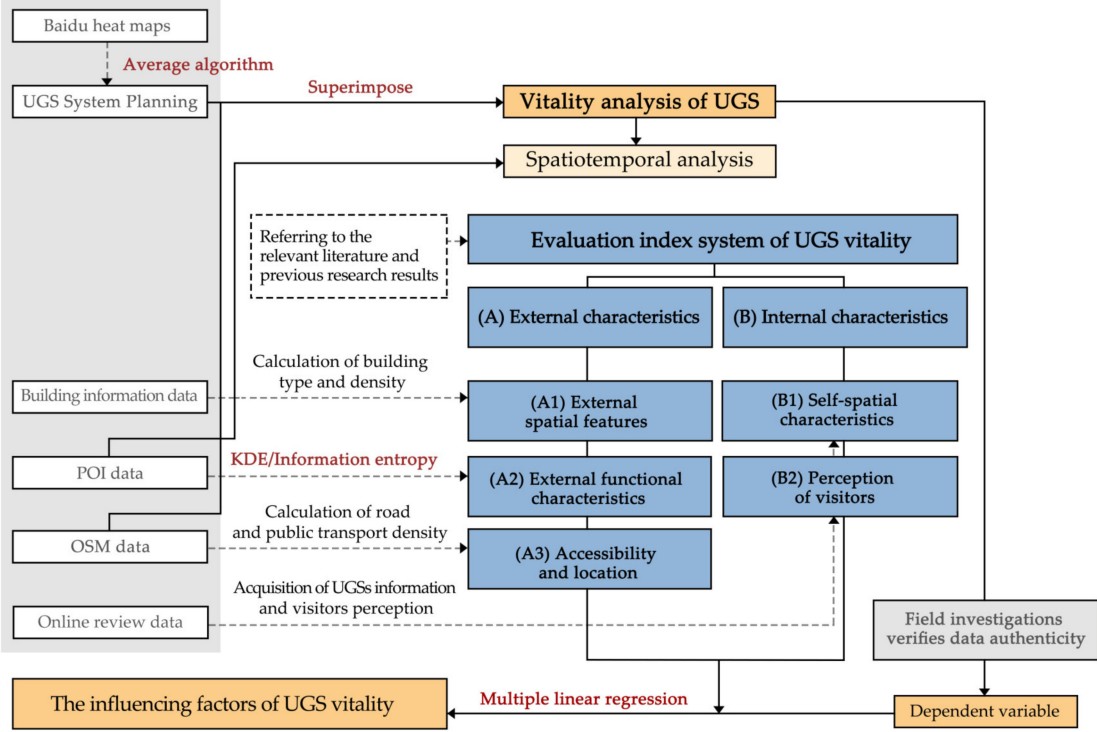

**Figure 2.** Identification process of the factors influencing UGS vitality.

*2.4. Methodology and Research Process*

2.4.1. Measurement of Spatial Vitality

Cities thrive because of people, and the aggregation of people in space can be regarded as the external expression of vitality [1,2,9]. Therefore, this study is based on Baidu heat maps, which reflect the spatial and temporal distribution of people as a means of indicating spatial vitality. The principle is based on the location information of mobile phone users, and according to this location, the distribution of crowd density, traffic flow and road conditions in the region and the associated spatiotemporal differences in the aggregation are displayed on the map in a visual way [15]. The closer the color is to red, the higher the relative population density, and the closer it is to green, the lower the population density. Due to the thermodynamic data being updated once every 15 min, related research often relies on the average algorithm to extract the average spatial vitality every hour [15,30]. The formula is as follows:

$$\overline{H} = \frac{\sum H_{i\mathrm{x}}}{n} \tag{1}$$

where $\overline{H}$ is the daily average population density of unit I, denoted by the number of people per unit space. $H_{i\mathrm{x}}$ is the population density of unit $i$ at time x, and $n$ is the time range of the data.

We obtained the weekly average value of regional spatial vitality by superimposing the Baidu heat maps concerning 7–24 h of work days and rest days (17 h of human daily activities) within the Third Ring Road of Chengdu [30,31]. We used the ArcGIS raster calculator and reclassified the results into 7 grades according to the Jenks natural discontinuity classification method; that is, the higher the vitality, the higher the grade. Subsequently, the green space distribution map of Chengdu Park Urban Green Space System Planning was superimposed on the Baidu heat maps data to obtain the spatial distribution of the average vitality in each UGS.

2.4.2. Construction of the Framework of the Factors Influencing UGS Spatial Vitality

For UGS spatial vitality, it is not sufficient to focus only on internal space. The urban environment outside UGS also has an impact on the UGS spatial vitality. Therefore, in this

study, we selected spatial vitality as the dependent variable. The independent variables are the "internal" and "external" characteristics that affect the vitality of UGSs that have commonly been used in previous studies. Thus, the main framework for the factors influencing UGS spatial vitality was constructed.

Among these factors, the external space of UGS covers public services, commercial services, residential areas and other facilities in the urban function that provide diverse spaces for the city. By analyzing the spatial differences in the degree of completeness, density and diversity of these various facilities, we can understand the spatial heterogeneity of vitality in the city, which can map and influence UGS, thus leading to differences in vitality [9,13,15,17,23,26,33]. Therefore, we chose the three indexes of (A1) external spatial features, (A2) external functional characteristics and (A3) accessibility and location as the evaluation elements pertaining to the "external" characteristics of spatial vitality, which basically cover all the contents of UGS external space, with the aim of fully understanding the factors influencing UGS external space vitality.

The internal dimensions of UGS, such as environmental quality, layout structure, functional area and infrastructure, are important indicators that reflect the degree of internal perfection of UGS. To analyze this content, the current situation of UGS can be understood in terms of its own spatial level. In addition, the perceptions, levels of satisfaction and behavioral activities exhibited by the crowd in the UGS internal environment can provide feedback regarding the park's vitality from the perspective of users [9,13,23,26,32]. Accordingly, with regard to the "internal" characteristics, we chose (B1) self-spatial characteristics and (B2) perceptions of visitors as indicators of the impression factors related to internal space vitality. The space quality can thus be combined organically with the subjective feelings of the crowd (Table 2).

**Table 2.** The influence index of UGS vitality.

| Type | | Index | Unit | Calculation Method |
|---|---|---|---|---|
| **Large Class** | **Medium Class** | | | |
| (A) External characteristics of UGS | (A1) External spatial features | (1) Peripheral development intensity | - | Ratio of the total building area to total land area within the green space service radius |
| | | (2) Surrounding building density | % | Building density within the green space service radius |
| | | (3) Proportion of surrounding residential land | % | Ratio of residential area to total land area within the green space service radius |
| | (A2) External functional characteristics | (4) Urban functional density | Count/km$^2$ | Density of various types of POI within the green space service radius |
| | | (5) Urban functional mixing degree | - | Mixing degree of functional facilities within the green space service radius |
| | (A3) Accessibility and location | (6) Density of public transport | Count/km$^2$ | Density of traffic facilities within the green space service radius |
| | | (7) Distance to nearest traffic stop | m | Distance to transportation facilities within the green space service radius |
| | | (8) Density of surrounding road network | m/km$^2$ | Road network density within the green space service radius |
| | | (9) Distance to the city center | m | Straight-line distance from the city center |

**Table 2.** *Cont.*

| Type | | Index | Unit | Calculation Method |
|---|---|---|---|---|
| **Large Class** | **Medium Class** | | | |
| (B) Internal characteristics of UGS | (B1) Self-spatial characteristics | (10) Area of green space | hm$^2$ | - |
| | | (11) Ratio of green space | % | Ratio of green area to total area |
| | | (12) Percentage of water areas | % | Ratio of water area to total area |
| | | (13) Number of entrances and exits | Count | - |
| | | | - | If activity facilities are planned, the value is 1; otherwise, it is 0 |
| | | (15) Availability of activity facilities | - | If parking lots are planned, the value is 1; otherwise, it is 0 |
| | | (16) Internal walking path density | m/km$^2$ | Density of walking paths in the green space |
| | (B2) Perception of visitors | (17) Comprehensive score of green space | Score | Visitors' ratings of the green space |
| | | (18) Number of green space reviews | Count | - |

2.4.3. Quantification of the External Influencing Factors Evaluation Index

To strengthen the relationship between external factors and UGSs, we took the spatial geographic center of UGSs as the center of the circle and delimited a service radius of 2 km for each UGS according to the requirements for the establishment of disaster prevention and buffer areas in Chinese cities and the literature on the division of service radii. We used the results to analyze the influence of urban space elements within the service radius on the vitality of UGSs [34,35]. At the same time, to better show the distribution of functional facilities around green space, we constructed uniform 100 m × 100 m space units during data visualization to realize the intuitive expression of data content and its placement in space.

1.   Analysis of indicators related to external spatial characteristics

External spatial features include peripheral development intensity, surrounding building density, and the proportion of three indicators of the surrounding residential land, which can express the correlation between UGS and the surrounding built environment space and is an important indicator of the influence of the form of the external space on the internal space. In addition to the surrounding building density, the other evaluation indicators have clear meanings and can be used directly.

2.   Analysis of indicators related to external functional characteristics

The source of urban vitality is the diversity of cities, and diversity is an important factor in promoting urban vitality [2]. The external function features include 2 indexes, urban functional density and urban functional mixing degree, which are used mainly to measure spatial functions and business forms around UGSs. This concept is relatively abstract. To facilitate later analysis, POI data should be quantified and spatially expressed first. Functional density is calculated by kernel density estimation (KDE) [36,37]. The formula is as follows:

$$f_{(s)} = \sum_{i=1}^{n} \frac{1}{h^2} k\left(\frac{s - c_i}{h}\right) \tag{2}$$

where $f_{(s)}$ is the kernel density calculation function in space $s$, $h$ is the distance attenuation threshold (bandwidth), $n$ is the number of elements with a distance from position $s$ that is less than or equal to $h$, $k$ is the spatial weight function, and $c_i$ is the core element.

The functional mixing degree was calculated by information entropy. The formula is as follows:

$$H_{(x)} = \sum_{i=1}^{n} P_i \log P_i \tag{3}$$

where $H_{(x)}$ represents the mixing degree of urban functions of land x and $P_i$ is the proportion of POIs of type $P$ in land x. The higher the value of $H_{(x)}$ is, the higher the functional mixing degree. $P_i$ represents the proportion of various POIs. Finally, according to whether the proportion of each function in the unit is more than 50%, we determine whether it is a single-functional area or a mixed-functional area.

3.    Analysis of indicators related to accessibility and location

Considering the travel types of visitors, there are 4 indicators: density of public transport, distance to nearest traffic stop, density of surrounding road network, and distance to the city center. This content reflects the size of the spatial resistance that people need to overcome to reach the destination [38]. The quantification of this evaluation index visually expresses POI data and road network data in ArcGIS through Formula (2).

### 2.4.4. Internal Characteristics of UGS

1.    Analysis of indicators related to self-spatial characteristics

Self-spatial characteristics are used to measure the environmental quality of green space, and there are 7 relevant evaluation indicators: area of green space, ratio of green space, percentage of water areas, number of entrances and exits, provision of parking lots, availability of activity facilities, and internal walking path density. Each indicator has clear semantics and can be directly expressed by relevant data.

2.    Analysis of indicators related to perceptions of visitors

The visitor satisfaction score from social data is one of the factors affecting travel choice as well as an important factor affecting the popularity of green space use [39]. There are 2 evaluation indicators: green satisfaction and the number of green space reviews. These indicators are calculated based on online review data; the comprehensive scoring index of green space takes the average of each sample data, and the number of green space comments reflects visitors' attention to each green space and the influence of green space through quantitative calculation.

### 2.4.5. Analysis of the Influencing Factors of Spatial Vitality

After model construction and data quantification, taking into account the different backgrounds of the internal and external environments and the constituent factors of each UGS, we screened out unnamed UGSs [23]. These spaces lack the support of visitors' perception data and cannot reflect visitors' real feelings about the space. Subsequently, we conducted correlation analysis using SPSS software to compare the quantified evaluation indexes with the aim of verifying their correlation. Finally, we used multiple linear regression to set up backward regression to gradually eliminate irrelevant indicators, analyze the contribution of each indicator to UGS vitality, and identify the key indicators that affect the popularity of UGS use [26,33]. The formula is as follows:

$$y = \beta_0 + \beta_1 * x_1 + \beta_2 * x_2 + \beta_n * x_n \tag{4}$$

where y is the dependent variable, $\beta_0$ is a constant term, $\beta_1$, $\beta_2$... $\beta_n$ are numbers of undetermined parameters, y is the dependent variable, and $x_1$, $x_2$... $x_n$ are independent variables to verify the impact of each index on UGS vitality.

## 3. Results
### 3.1. Vitality Analysis of UGS
3.1.1. The Spatial and Temporal Distribution of Vitality

In this study, we obtained a spatial distribution map of the average vitality of UGSs in Chengdu. The results cover two types of green space: park green space (urban park, community park, square, zoo, etc.) and regional green space (production green space, wetland park, etc.). The results include 38 visible UGS patches, but only 22 named UGSs contain evaluation data that can be used. The results were assigned different colors on a

scale ranging from 1–7; the higher the score is, the redder the color and the more intensive the use (Figure 3). The findings indicated that the average vitality value of UGSs in Chengdu was 2.868 and that the overall vitality was low. Simultaneously, for vitality levels 1–7, the vitality of UGSs in Chengdu reached values of 29.73%, 27.03%, 13.51%, 13.51%, 2.70%, 10.81%, and 2.70%, respectively. In terms of the spatial distribution of UGS vitality, we identified the following characteristics:

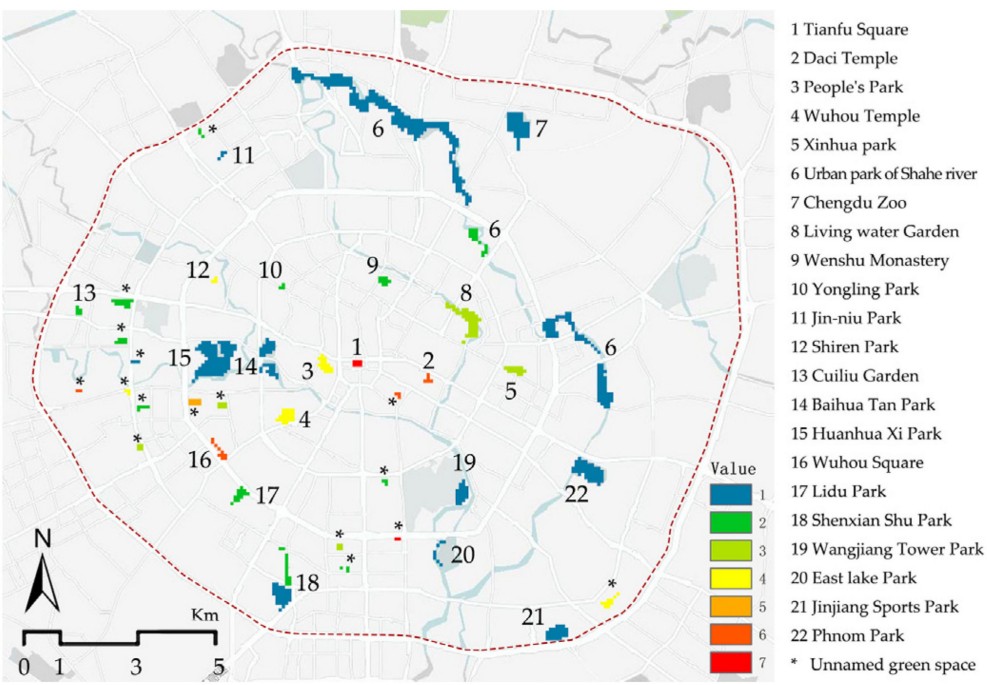

**Figure 3.** Spatial distribution of UGS vitality within the Third Ring Road of Chengdu.

1.　　UGS vitality shows high centrality

　　UGS areas with higher vitality values were located mainly in the center of Chengdu, showing centrality and a decreasing vitality distribution trend from the center outward. For example, Tianfu Square (1) and People's Park (3) are located in the center of the research area, and Daci Temple (2) and Wuhou Temple (4) are located near the center and other UGS. Although these green spaces are small in area, they show spatial distribution characteristics of very high vitality against the background of resource abundance and dense population in the urban center.

2.　　The vitality of UGSs with large areas is relatively low

　　Chengdu Zoo (7), Huanhua Xi Park (15), Phnom Park (22) and other parks with large areas have relatively low vitality. As "old brand" parks in the downtown area of Chengdu, these parks have large areas, diverse internal functions and abundant vegetation, forming good green spaces. However, due to the large green areas of the parks, the dispersion of spatial functions and an excessive number of inaccessible areas, the density of crowd gathering may be relatively low, leading to the phenomenon of low average park vitality, which needs to be further verified in the regression analysis.

### 3.1.2. Spatiotemporal Analysis of Vitality and Related Indicators

　　Through the spatial vitality data for weekdays and weekends, we calculated the space usage proportion of each selected UGS in different periods. We further divided the spatial vitality according to high (6–7 levels), medium (3–5 levels) and low (1–2 levels) values (Figure 4). The figure shows that although the vitality of UGS on weekends is higher than that on weekdays, the change trends of the two are relatively similar. Among them, the average value of vitality on weekdays and weekends peaked at 13–16 p.m. and then

gradually decreased, reaching its lowest point at approximately 11 p.m. Simultaneously, the low-value dashed line of weekday and weekend vitality shows change characteristics of increase → decrease → increase, but there are differences in terms of the period of decrease. In terms of the high-value dashed line, the high value of work days and rest days appeared for Tianfu Square (1), Daci Temple (2), and People's Park (3), showing strong clusters of vitality.

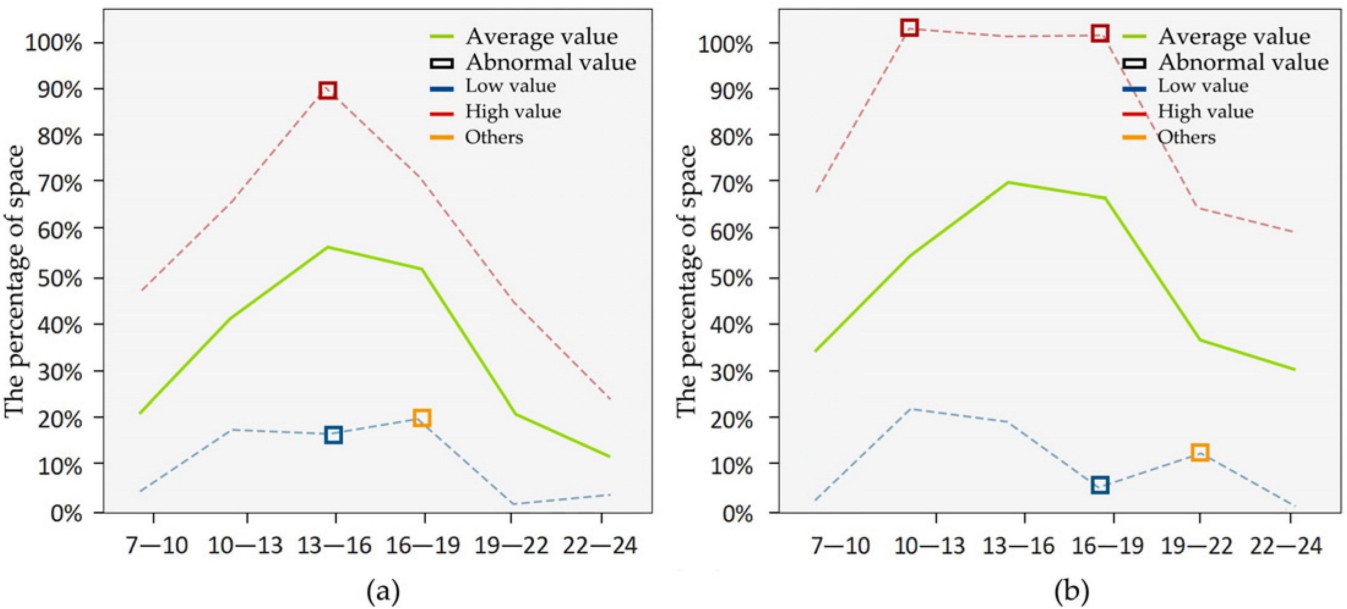

**Figure 4.** Proportion of space with vitality above level 3 on weekdays (**a**) and weekends (**b**).

Various indicators that measure the popularity of use also show different spatiotemporal heterogeneity. Among the external functional features, 131,259 POIs were obtained through screening, with an average data density of 1078/km². The data density is divided into seven levels according to the degree of aggregation (the red color represents higher density), and the external functional density analysis chart of UGSs in Chengdu is obtained (Figure 5).

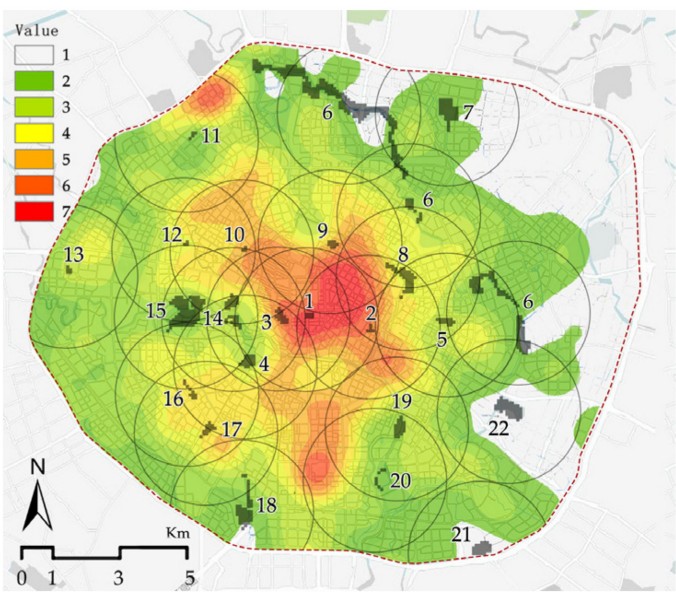

**Figure 5.** Spatial heterogeneity of functional density.

First, the functional density of the Third Ring Road in Chengdu shows spatial distribution characteristics of lower density in the east and higher density in the west and decreases from the center outward. This pattern indicates that the spatial distribution of functional density still shows high centrality; that is, the urban center is dense and has a high degree of aggregation. In addition, a small number of independent high-vitality zone clusters appear in facilities located on the periphery of the center. This heterogeneity creates considerable differences in the functional density of the UGS service radius in Chengdu, and the density value interval indicates a density quantity difference of 2280/km$^2$. Second, the spatial and temporal heterogeneity of functional density is also reflected in high-concentration areas above level 5. For example, People's Park (3), which has the highest proportion of level 5 or above, accounts for 7.12%, while Chengdu Zoo (7) accounts for only 0.56%, indicating a significant difference in the number of spaces.

The spatial distribution of the functional mixing degree also shows spatial distribution characteristics of lower density in the east and higher density in the west and decreases from the center outward. In terms of spatial proportion, there are few differences between single-functional areas, mixed-functional areas and no-data areas, which are more balanced in the urban space (Figure 6). Mixed-functional areas contain a concentration of multiple urban functions. UGSs near the city center, such as People's Park (3) and Wuhou Temple (4), account for more than 30%. At the same time, single-functional UGS areas in the city center account for more than 50%, which indicates that a certain type of functional space in this region has strong aggregation and is distributed in clusters. However, Chengdu Zoo (7) and Phnom Park (22), which are located close to the edge of the research area, have a small proportion of single-functional areas and mixed-functional areas due to the associated amount of data.

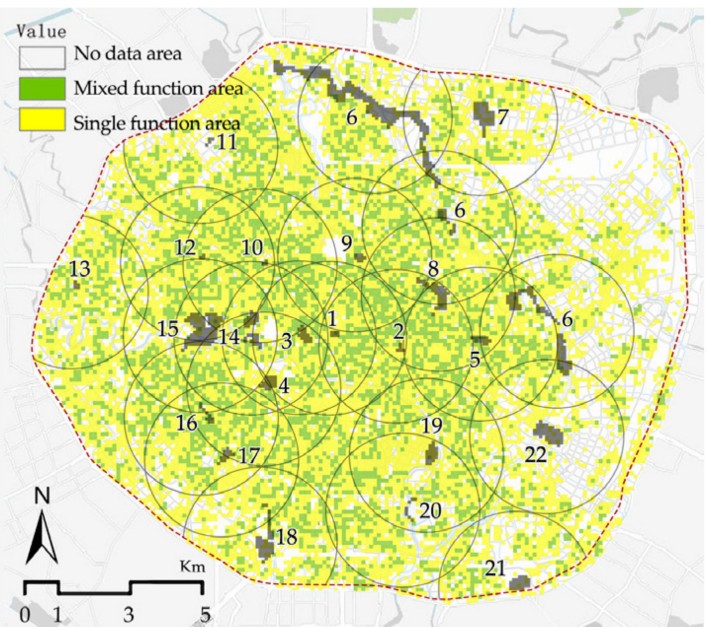

**Figure 6.** Spatial heterogeneity of the function mixing degree.

### 3.2. Analysis of the Factors Influencing UGS Vitality

Through the test of a multiple linear regression model, the contribution degree of each index to the vitality of UGSs can be distinguished. We gradually removed irrelevant influencing factors by a stepwise regression model and obtained 13 stepwise regression models. Except for models 1–4, all stepwise analysis results passed the significance test. However, the variance inflation factor (VIF) value in most models was greater than five, indicating collinearity. Therefore, the 13th model with all indicators was selected as the final evaluation result (Table 3). In the final results, the R2 and adjusted R2 were 0.922 and 0.882, respectively, showing a good degree of model fit and passing the F test (F = 23.491,

P = 0.000 < 0.05), thus indicating that the seven indicators explained 92.2% of the influence of UGS vitality.

**Table 3.** Results of the regression analysis.

| | Unstandardized Coefficients | | Standardized Coefficients | $t$ | P | VIF | $R^2$ | Adjusted $R^2$ | DW | $F$ |
|---|---|---|---|---|---|---|---|---|---|---|
| | $B$ | Standard Error | Beta | | | | | | | |
| (Constant) | 5.537 | 7.03 | | 0.788 | 0.444 | | | | | |
| (2) Surrounding building density | −0.077 | 0.042 | −0.227 | −1.84 | 0.087 | 2.717 | | | | |
| (4) Urban functional density | 0.002 | 0.001 | 0.454 | 3.133 | 0.007 | 3.741 | | | | |
| (7) Distance to nearest traffic stop | 0.005 | 0.001 | 0.445 | 3.401 | 0.004 | 3.059 | | | | |
| (8) Density of surrounding road network | 2.113 | 0.522 | 0.506 | 4.045 | 0.001 | 2.788 | 0.922 | 0.882 | 2.355 | 23.491 |
| (11) Ratio of green space | −0.057 | 0.014 | −0.463 | −4.03 | 0.001 | 2.36 | | | | |
| (14) Provision of parking lots | −2.111 | 0.556 | −0.327 | −3.798 | 0.002 | 1.322 | | | | |
| (17) Comprehensive score of green space | 3.225 | 1.222 | 0.276 | 2.639 | 0.019 | 1.95 | | | | |

### 3.2.1. Analysis Results of External Influencing Factors

According to multiple linear regression analysis, (2) surrounding building density, (4) urban functional density, (7) distance to nearest traffic stop, and (8) density of surrounding road network have a certain impact on spatial vitality. Except for (2) surrounding building density, the other indicators all show a positive influence on spatial vitality. Among them, (8) density of surrounding road network has a significant impact on the popularity of use, reflecting the impact of traffic factors on space in the development of the city. Additionally, (4) urban functional density expresses the influence of urban functional facilities on the popularity of use. According to the regression analysis results, urban functional density is positively correlated with the popularity of UGS use. In other words, the better the service facilities surrounding UGSs are, the more likely it is that the number of users of green space will increase to a certain extent, indicating that diverse urban spaces will also increase the willingness of urban residents to use green space to a certain extent.

### 3.2.2. Analysis Results of External Influencing Factors

According to the regression analysis, (11) the ratio of green space and (14) the provision of parking lots exhibited negative correlations with spatial vitality, whereas (17) the comprehensive score of green space did not. These findings, as the subjective satisfaction score of visitors with UGSs, can provide a reference for subsequent visitors to make travel decisions. According to the regression model results, (17) the comprehensive score of green space has a strong influence, on spatial vitality, reflecting the choices and references made by modern urban residents before venturing out for recreation.

## 4. Discussion

### 4.1. Introducing Baidu Heat Maps to Complete the Spatial Vitality Measurement

The use of urban space reflects the degree of support and development of the city for human life [4,40]. The concept of vitality has often been used in previous studies to evaluate urban space. How to judge and quantitatively express spatial vitality has long been a focus and challenge in research in this field [6,9,18,33,41]. Related studies have often identified urban vitality through superimposition analysis and evaluation of multiple urban spatial elements. However, are the coverage degree and expressiveness of vitality elements complete in these methods? Which method is used for scientific and reasonable superimposition? Further verification is needed.

People are the main body of urban development and the driving force of spatial prosperity [42,43]. Therefore, we chose Baidu heat maps, a spatial type of big data that directly reflects the spatiotemporal aggregation of people, as the data sources for spatial vitality to represent the external expression of vitality in UGSs. In this study, the spatial vitality of each hour was calculated by means of the mean value algorithm, and the measurement of the vitality of UGSs in Chengdu was then completed by means of ArcGIS

data visualization and cutting. The results reflect the spatial differentiation characteristics based on real-time population aggregation and the real use of UGSs.

At present, regardless of the method that is used, almost all studies in this field use reference big data, which is a very new data source. However, there are still some problems with using this kind of data. First, most spatial big data are generated by users carrying smartphones. Even though the number of elderly netizens aged 60 years and above in China reached 119 million in 2022, 56.8% of the elderly group and the younger group cannot be identified in this kind of data, so there is a lack of data on UGS spatial and temporal experience for these age groups [44]. In response to this problem, we selected People's Park, which has exhibited outstanding vitality performance at various times according to the spatial vitality data, and we conducted field research on 27 (weekdays) and 29 (weekends) Jan 2019. Using a combination of fixed-point observation and interviews, we verified the authenticity of the spatial vitality data and online review data. In the actual survey, the average spatial vitality value was 2.937 after conversion into seven levels of classification, which was basically consistent with the online spatial vitality data. However, because many elderly people exercise in UGS in the morning, online data lacks some content. Although research can address this issue, it remains a problem that must be solved if the integration of online and offline data regarding spatial vitality is to be realized. Second, the construction of tall buildings and the expansion of underground space provide rich vertical space for the city, but spatial big data are mostly in two-dimensional form, which leads to a certain degree of accumulation of these vertical space data after flattening in the plane space, resulting in a low longitude of the data. Future research can consider these two problems and their solutions to further improve the accuracy and authenticity of the data.

*4.2. Analysis of the Influence Index of UGS Vitality*

The determination of indicators in the influencing factor model has a great influence on the analysis of spatial vitality. The content and quantity should cover all urban space elements; otherwise, the analysis can lead to an unclear use of vitality sources and be unable to identify a direction for optimization. Compared with studies that have focused only on the physical space inside and outside UGSs, this study integrates visitors' perceptions into the internal feature elements and considers human-oriented urban development characteristics. From this perspective, (17) comprehensive score of green space shows a strong positive correlation with spatial vitality, which is consistent with other research results. According to relevant studies, destination satisfaction and rating have a significant positive impact on visitors' behavioral intention, and satisfaction moderates the relationships between destination attraction factors, destination image and intention to revisit, playing an important mediating role [45,46]. This finding indicates that in modern life, online comment information serves as an important reference for people with respect to making decisions regarding their destinations [47]. Second, (11) the ratio of green space showed a weak negative impact, which was consistent with the results of a study in Shanghai, China [48]. In places with a high green rate, the physical space available to support various activities is relatively small, which leads to a large number of inaccessible areas in the space. This hinders the development of space activities, making the overall space utilization rate low, which affects the spatial vitality level.

There are differences between the results of this study and those of previous studies. First, although (4) urban functional density has a positive impact on the popularity of UGS use, its influence is weak. Additionally, (5) urban functional mixing degree has a low correlation and is excluded from the regression analysis. This finding is somewhat different from the concept of "rich and diverse urban space is the source of vitality" proposed by Jacobs [2]. At the same time, studies have found that high-density and mixed land use are closely related to neighborhood vitality and directly affect spatial vitality [49]. The reason may be that the superposition of multiple elements weakens the influence of urban external functional features on space. Second, (14) provision of parking lots has a particularly prominent negative impact on the vitality of UGSs, which is contrary to other accessibility

indicators and relevant research conclusions in this study. An analysis of the reasons indicates that this factor is similar to the green land rate index in this study, which may invade the internal space of green land to some extent. At the same time, the negative impact of this factor significantly reflects differences in the travel modes of urban residents to green space [50]. The textual analysis of online comment data reveals that most residents choose public transportation or walking to reach UGSs.

Determining the influencing factors of spatial vitality is a complex and large task that requires constant elaboration and verification of model indicators. The expression of urban external functional characteristics and their inclusion types in this study is still not comprehensive. Future research can further refine and classify urban functional services to reflect the diversity of cities. Moreover, this study delimits the service radius of UGSs according to relevant norms and service radii. This division method is relatively convenient, but it is inadequate for more refined and accurate urban development requirements. Future research can use isochronal, accessibility quantification and other methods to divide the service radius according to the requirements of the urban life circle, but attention should be given to the rationality and scientific basis of the division [51]. At the same time, this study takes UGSs in Chengdu as the object. In terms of usage and related indicators, studies have found that these indicators have obvious spatiotemporal heterogeneity; thus, whether this heterogeneous background will interfere with the influencing factors of usage still needs to be further explored. Future research can seek to classify urban green space according to the level of vitality value and surrounding environment in further detail and can explore the differences in the associated influencing factors under object classification.

### 4.3. Inspiration for UGS Optimization Guided by Spatial Vitality Enhancement

In the future development of UGSs, to improve space vitality and meet the needs of visiting urban residents, high-density and high-quality UGSs are particularly important. Therefore, based on the analysis results concerning the factors influencing spatial vitality and taking into account the relevant policies issued by the government concerning UGSs, this study proposes the following four strategies to optimize the spatial vitality of UGSs.

First, UGS accessibility should be improved. Under the urban development requirements of the "15 min life circle" proposed by China and "seeing the green from 300 m and seeing the park from 500 m" proposed by Chengdu [52], the traffic cost to reach UGS should be reduced to the greatest extent possible, and the population served by green space should be considered to avoid overcrowding. The results of this study show that the closer a UGS area is to the city center, the greater the human flow, the more complete the urban infrastructure, the better the UGS accessibility and the more recreational opportunities there are. However, in such urban core areas, the construction of UGSs has been basically completed, leaving little possibility for the development of new areas. Therefore, in this kind of urban space, the development of new UGSs should make full use of underutilized areas of the city, strengthen the construction of "pocket parks" and "street-corner gardens", and improve the accessibility of UGSs and the utilization of urban space so that urban residents can enjoy UGS services within 15 min on foot. At the same time, for existing green spaces, the openness of these areas and the density of surrounding public transport should be further improved to facilitate better access.

Second, the connectivity between UGSs and the surrounding environment should be strengthened. The outer space development and functional distribution of UGS also have a certain influence. To optimize and improve these features, population characteristics, land use types, socioeconomic development and other aspects of the service radius of UGSs should be taken into account from the perspective of top-level urban planning to realize orderly spatial connectivity. At the same time, the openness of UGSs and their connectivity with the surrounding environment should be strengthened. For example, Chengdu proposed the idea of "park city" scene construction, requiring UGS construction to be integrated into the consumption scene to further diversify green space and ultimately promote the generation of vitality through consumption [53].

Third, attention should be given to improving the attractiveness of UGSs. The negative correlation between the green space rate and the two indexes of parking lots in this study indicates that sightseeing space is important for UGS. Therefore, to improve the attractiveness, the internal characteristics of green space that can attract visitors, such as landscape resources, the richness of activity space, and the proportion of impervious surface to green space, should first be ensured. At the same time, to ensure that UGSs can attract visitors, it is necessary to optimize the organization of space and tour routes, activities and facilities based on human needs to enable UGSs to better serve people.

Finally, we should focus on user experience and establish a feedback mechanism. UGS serves people. Therefore, in the process of planning, designing or optimizing green space, visitors' feelings and demands for green space should be considered, and network evaluation data should be reasonably used to determine what hinders green space usage. These data and the network comment data in this study indicate that there are some problems in UGSs in Chengdu, such as inadequate maintenance and management, unreasonable space division and imperfect service facilities. A feedback mechanism can be constructed in the process of green space renewal, which can improve visitors' satisfaction and feelings about UGS and provide ideas and guidance for space renewal.

## 5. Conclusions

Spatial vitality is an important index for measuring the utilization of UGS. In this study, spatial big data such as Baidu heat maps, POIs, OSM and online comments were used to measure the vitality of UGSs in Chengdu, and five influencing factors of the vitality of UGSs were analyzed: external spatial characteristics, external functional characteristics, accessibility of green space, self-spatial characteristics and recreation satisfaction.

In contrast to previous studies, in this study, we constructed a vitality analysis model that combined the internal and external characteristics of UGSs. Our model integrates POI data represented by city features (functional density, mixed degree) and network review data represented by leisure satisfaction (green comprehensive score and comments) as influencing factors to further strengthen the organic connection of UGSs and external space with visitors and to optimize the regression model.

The results of this study reveal that green space and urban resources in Chengdu have characteristics of high central aggregation in spatial vitality, POI density and other evaluation indicators. In other words, higher road network density and closer transportation facilities help visitors better reach parks. Additionally, the surrounding functional density has a certain promotional effect on spatial vitality, reflecting the influence of the diversified city. The regression results for surrounding building density, ratio of green space and provision of parking lots show that a reasonable and appropriate spatial and functional layout can contribute to the development of green space. In addition, the high rating of visitors for UGSs has a significant positive influence on spatial vitality. Establishing a humanized space with a feedback mechanism can enable UGSs to better meet people's needs and improve green spatial vitality.

**Author Contributions:** Conceptualization, Q.D. and J.C.; methodology, Q.D.; software, Q.D.; validation, J.C., S.C. and P.H.; investigation, Q.D. and P.H.; resources, X.C.; data curation, S.C.; writing—original draft preparation, Q.D., J.C., S.C. and P.H.; writing—review and editing, Q.D., J.C. and X.C.; visualization, Q.D.; supervision, J.C.; project administration, X.C. All authors have read and agreed to the published version of the manuscript.

**Funding:** This research received no external funding.

**Data Availability Statement:** Not applicable.

**Conflicts of Interest:** The authors declare no conflict of interest.

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
