# Peer review of "Spatiotemporal Analysis of Urban Green Spatial Vitality and the Corresponding Influencing Factors: A Case Study of Chengdu, China"

_land, doi:10.3390/land11101820_

Round 1
Reviewer 1 Report
Dear authors,
Firstly, I would like to thank you for your work in this field.
You did an excellent job!
But for the current version, I have some comments for it, not many but essential.
For revision details, please see the followings:
1. The research questions are implicitly provided within the Introduction section. The authors should provide them directly at the end of the section. Each research question should correspond to one specific research objective.
2. Figure 1. Technical Route should continue the problem orientation in the introduction section. It should also be emphasized what specific problems were solved at that stage.
3. The name of Figure 4 is written in Chinese, which is difficult to understand. Please reconsider the language use.
4. The Discussion section should be arranged to follow the original order of Research Questions. The explanations should be clear to indicate which key findings answer which question. For example 3.1.2. The analysis results of Spatial and temporal heterogeneity of use heat and related indicators should be added to the discussion content, and the Spatial differences of vitality brought by such heterogeneity should be discussed.
5. Please consider discussing how the research findings compare with the current urban paradigms: 15-minute city, mixed-use development sustainable development, etc.
6. Please rethink the use of phrases and language style, e.g., development of urbanization = urbanization
Again, I have a highly comment on your contribution.
The upon revision recommendations are helping this manuscript be more readable for readers.
Author Response
Point 1: The research questions are implicitly provided within the Introduction section. The authors should provide them directly at the end of the section. Each research question should correspond to one specific research objective.
Response 1: Thank you for your constructive comment. According to your suggestions, we have provided a detailed explanation in the last paragraph of the Introduction, clearly describing the problems involved in the research and the research objectives arising from these problems.
Point 2: Figure 1. Technical Route should continue the problem orientation in the introduction section. It should also be emphasized what specific problems were solved at that stage.
Response 2: Thank you. According to the expression in "Figure 1. Technical Route" you mentioned, we made adjustments to ensure that the content was in line with the full text; we clarified the content to be expressed and the problems to be solved in each stage.
Point 3: The name of Figure 4 is written in Chinese, which is difficult to understand. Please reconsider the language use.
Response 3: Thank you for your suggestion. We have made adjustments to similar issues in the full text to ensure the legibility and standardization of the article.
Point 4: The Discussion section should be arranged to follow the original order of Research Questions. The explanations should be clear to indicate which key findings answer which question. For example 3.1.2. The analysis results of Spatial and temporal heterogeneity of use heat and related indicators should be added to the discussion content, and the Spatial differences of vitality brought by such heterogeneity should be discussed.
Response 4: Thank you for your constructive comment. The analysis content in "3.1.2 Spatial-temporal Analysis of Vitality and Related Indicators" has been elaborated in the Results. In view of the corresponding problems you mentioned before and after, we have added the discussion of this section to the last part of "4.2. Analysis of the Influence Index of UGS Vitality", which expounds the influence of spatiotemporal heterogeneity on the regression analysis results. The future optimization direction of this content is proposed.
Point 5: Please consider discussing how the research findings compare with the current urban paradigms: 15-minute city, mixed-use development sustainable development, etc.
Response 5: Thank you for your suggestion. We strongly agree that "15-minute city", "mixed-use development Sustainable Development" and other contents are indeed hotspots in recent urban studies, which have very high practical importance. This is explained in "4.3. Inspiration for UGS Optimization Guided by Space Vitality Enhancement", but not completely. Based on your comments, we have further adjusted the content to strengthen its relevance to the current hotspots you mentioned.
Point 6: Please rethink the use of phrases and language style, e.g., development of urbanization = urbanization.
Response 6: Thank you. To address this comment, we have carried out a comprehensive review of the expression of English phrases and the spelling.

Reviewer 2 Report
Title: Confusing part is, “…Green Space Use Heat…”
Abstract: Confusing part is “…... In this study, use heat was regarded……”
Lines 17 and elsewhere: Get rid of all word hyphen breaks by using Home>Paragraph>Line and Page Breaks>Check “don’t hyphenate”
Figure 1 caption: Use a self-explanatory text
Figure 2 caption: 3nd?
Equation 1: What is unit of H? Usually, n is written italic in equations.
Equation 4: Why the variables do not use subscripts as in text?
Figure 4 caption: Chinese vs. English
Figure 7: Label x-axis too
Author Response
Point 1: Confusing part
Title: Confusing part is, “…Green Space Use Heat…”
Abstract: Confusing part is “…... In this study, use heat was regarded……”
Response 1: Thank you for your constructive comment. The concept of "use heat" is truly difficult to understand. We have changed the term related to the topic to "spatial vitality". In addition, in the summary and 2.1, we explain that "the heat value contained in Baidu Heatmap was used as the external representation of spatial vitality". In this way, the spatial vitality proposed in this paper is explained so that it is easier to understand.
Point 2: Other existing problems
Lines 17 and elsewhere: Get rid of all word hyphen breaks by using Home>Paragraph>Line and Page Breaks>Check “don’t hyphenate”
Figure 1 caption: Use a self-explanatory text
Figure 2 caption: 3nd?
Equation 1: What is unit of H? Usually, n is written italic in equations.
Equation 4: Why the variables do not use subscripts as in text?
Figure 4 caption: Chinese vs. English
Figure 7: Label x-axis too
Response 2: Thank you. The format, figure name, legend and related normative content you mentioned have been further improved and supplemented in the revision. Based on these changes, the normative expression of other related content has been checked.

Reviewer 3 Report
The paper presents a study on urban green space use frequency and its influencing factors in Chengdu, China. The topic is interesting and important. However, I would like to suggest major revisions to the manuscript in order to make it suitable for publication. My comments and suggestions are as follows:
1. The term “use heat” in the manuscript is confusing. Literally, it means to use energy, which does not correspond to what the authors want to mean in this manuscript. I suggest using another term that is more direct, appropriate, and common in the literature, for example, “use frequency” as already mentioned in the manuscript.
2. The authors used many indicators and calculation methods. However, the presentation of methods and results is not straightforward enough nor well organized for the readers to follow easily what indicators and data sources were used, how the data were treated, what are the main results, etc. Concrete suggestions are as follows:
(1) The idea of using Figure 1 to show the global framework of the research design is good. However, the Figure needs to be significantly improved:
For example, it needs to distinguish between methods, indicators, data sources, etc. by using the same color for each category and different colors for different categories.
Then, what does “Construction of evaluation index system of urban green space utilization heat” mean? If these external and internal factors are variables for the regression analysis below, it would be better to put them below the box “Factors influencing the utilization…”.
Meanwhile, the framework can integrate more information to be able to deliver a whole picture to the readers about the research design, for example, the time periods of the analysis (weekdays, weekend days, years, months, …).
(2) Table A1 and A2 should be integrated into the text rather than into the appendix. Especially, Table 1 presents a quite clear structure of the indicators (dimensions A/B, A1-A3, B1-B2, indicators 1-18). The table is important for the readers to understand the following calculation and results.
Figure 1 should present the same structure of the internal and external factors as Table 1, that is to say, showing the dimensions A1-A3 and B1-B2 instead of a part of the indicators (“Area of green space”, …).
Figure 7 is in redundancy with Table A2. It would be better to keep only Table A2.
(3) The authors need to justify their methodology and selection of indicators in Section 2.1 research design. Why these internal and external characteristics are influencing factors of urban green space use frequency? What has been done in the literature? The authors present in the manuscript what they did without explaining why it would be meaningful to do these analyses.
(4) Section 2.3 Data sources: It is not easy to follow during which period the data for each indicator were collected. For example, from the saying: “the data from 2018 were selected to avoid the impact of COVID-19 on travel”, it is not clear what period is used. I would suggest adding a table clearly showing all the data sources.
3. About the research objective, the authors claim in P3, L96-98 “we analyze the influencing factors of UGS use heat and the key points of high-heat UGS planning and construction to ensure that it is aligned with the public’s behavioral habits and psychological needs. The purpose is to improve the utilization rate and space vitality of UGS.” The statement is a little too vague and fuzzy. How the analysis of influencing factors will ensure that the UGS planning is aligned with the public’s behavior and needs? How the study of frequency patterns and influencing factors will contribute to improving the utilization and space vitality of UGS? The authors need to be precise and complete their logical chains between the objectives and the methodology. They also need to come back after showing the results and verify with the readers how their results respond to their objectives. There is some discussion on this type in Section 4.3, but the discussion on the contributions of the results to their objectives (to improve UGS utilization, planning, and vitality) should be strengthened.
4. It seems that the authors relied only on online data sources. Some field investigations would be helpful and even necessary for this study because, just like the authors said, the Baidu heat map is dependent on smartphone users, and the visitors’ comments on the website are not necessarily representative. It is totally possible to carry out some field interviews at the scale of this study. There might be two suggestions:
(1) To carry out some field control investigations to verify the effect of the Baidu heat map in the identification of UGS use frequency.
(2) To carry out some field investigations/interviews to distinguish the user groups in order to provide more meaningful conclusions and practical recommendations.
5. The English editing and typos errors need to be improved.
Author Response
Response to Reviewer 3’s Comments
The paper presents a study on urban green space use frequency and its influencing factors in Chengdu, China. The topic is interesting and important. However, I would like to suggest major revisions to the manuscript in order to make it suitable for publication. My comments and suggestions are as follows:
- The term “use heat” in the manuscript is confusing. Literally, it means to use energy, which does not correspond to what the authors want to mean in this manuscript. I suggest using another term that is more direct, appropriate, and common in the literature, for example, “use frequency” as already mentioned in the manuscript.
Response 1: Thank you for your constructive comment. The concept of "use heat" is truly difficult to understand. However, if we were to choose the term "frequency of use" instead, we would require more research on different user groups to explain the concept of "frequency" clearly. Therefore, we have changed the term we use to refer to this topic to "spatial vitality". In addition, in the summary as well as in 2.1 and 2.4, we explain that "Cities thrive because of people, and the aggregation of people in space can be regarded as the external expression of vitality……the heat value obtained from Baidu heat maps was used as an external representation of spatial vitality". In this way, the notion of spatial vitality proposed in this paper is explained in a manner that is easier to understand.
- The authors used many indicators and calculation methods. However, the presentation of methods and results is not straightforward enough nor well organized for the readers to follow easily what indicators and data sources were used, how the data were treated, what are the main results, etc. Concrete suggestions are as follows:
(1) The idea of using Figure 1 to show the global framework of the research design is good. However, the Figure needs to be significantly improved:
For example, it needs to distinguish between methods, indicators, data sources, etc. by using the same color for each category and different colors for different categories.
Then, what does “Construction of evaluation index system of urban green space utilization heat” mean? If these external and internal factors are variables for the regression analysis below, it would be better to put them below the box “Factors influencing the utilization…”.
Meanwhile, the framework can integrate more information to be able to deliver a whole picture to the readers about the research design, for example, the time periods of the analysis (weekdays, weekend days, years, months, …).
(2) Table A1 and A2 should be integrated into the text rather than into the appendix. Especially, Table 1 presents a quite clear structure of the indicators (dimensions A/B, A1-A3, B1-B2, indicators 1-18). The table is important for the readers to understand the following calculation and results.
Figure 1 should present the same structure of the internal and external factors as Table 1, that is to say, showing the dimensions A1-A3 and B1-B2 instead of a part of the indicators (“Area of green space”, …).
Figure 7 is in redundancy with Table A2. It would be better to keep only Table A2.
Response 2: Thank you for your comments in (1) and (2). In response to your suggestions, we have adjusted the positions of Table 1 and Table 2 and deleted the contents of Figure 7. In addition, due to the problem of the location of Figure 1, the data sources and methods were proposed following Figure 1, and so we have moved the current "Figure 1" to the end of “2. Materials and Methods”, which allows us to illustrate all the methods, indicators, data sources and other contents mentioned in this chapter clearly with the aim of ensuring that the logical structure of this study can be fully represented.
(3) The authors need to justify their methodology and selection of indicators in Section 2.1 research design. Why these internal and external characteristics are influencing factors of urban green space use frequency? What has been done in the literature? The authors present in the manuscript what they did without explaining why it would be meaningful to do these analyses.
Response 2.3:Due to the adjustment of the location of the original Section 2.1, we have placed our discussion of the selection of relevant indicators in the current Section 2.3.2 to make the selection of indicators more reliable. Regarding the content of the original research design, we put this discussion at the end of "2. Materials and Methods" (Section 2.4), which primarily focuses on explicating the whole research framework in terms of data, methods, processes, etc., so that readers can obtain a clearer understanding of the paper’s structure.
(4) Section 2.3 Data sources: It is not easy to follow during which period the data for each indicator were collected. For example, from the saying: “the data from 2018 were selected to avoid the impact of COVID-19 on travel”, it is not clear what period is used. I would suggest adding a table clearly showing all the data sources.
Response 4: Thank you. For this reason, we have included a table to explain the time of each data expression and illustrate all data sources clearly.
- About the research objective, the authors claim in P3, L96-98 “we analyze the influencing factors of UGS use heat and the key points of high-heat UGS planning and construction to ensure that it is aligned with the public’s behavioral habits and psychological needs. The purpose is to improve the utilization rate and space vitality of UGS.” The statement is a little too vague and fuzzy. How the analysis of influencing factors will ensure that the UGS planning is aligned with the public’s behavior and needs? How the study of frequency patterns and influencing factors will contribute to improving the utilization and space vitality of UGS? The authors need to be precise and complete their logical chains between the objectives and the methodology. They also need to come back after showing the results and verify with the readers how their results respond to their objectives. There is some discussion on this type in Section 4.3, but the discussion on the contributions of the results to their objectives (to improve UGS utilization, planning, and vitality) should be strengthened.
Response 5: Thank you for your suggestion. To address the problem of vague expression and unclear logic of “P3, L96-98” that you mentioned, we have split the relevant content. In this context, we put the content pertaining to the "goal" at the end of "1. Introduction" and moved the content regarding the research framework to "2.4 Research Framework". Simultaneously, we strengthened the goal orientation in Section 4.3 as well as the connection with the content of the previous article. This alteration makes the chain of reasoning of the whole article clearer and allows us to explain the relationships among problems, objectives, methods and results more fully.
- It seems that the authors relied only on online data sources. Some field investigations would be helpful and even necessary for this study because, just like the authors said, the Baidu heat map is dependent on smartphone users, and the visitors’ comments on the website are not necessarily representative. It is totally possible to carry out some field interviews at the scale of this study. There might be two suggestions:
(1) To carry out some field control investigations to verify the effect of the Baidu heat map in the identification of UGS use frequency.
(2) To carry out some field investigations/interviews to distinguish the user groups in order to provide more meaningful conclusions and practical recommendations.
Response 6: Thank you for your suggestion. With respect to the on-site comparison survey and interview you mentioned, we selected the People's Park, which exhibited outstanding data performance at the beginning of the study, and we conducted a two-day on-site survey (on both working days and rest days). In this context, we focused on elderly individuals' feelings regarding and satisfaction with the park. The survey results are mostly consistent with the research data, but we did not explain them in the text. After taking your suggestion into account, we compared the relevant research results with the data in the text to verify the applicability of online data.
In other respects, regarding the problem of distinguishing user groups that you mentioned in (2), due to the limitations of space associated with this study, it is impossible to represent various types of user groups completely. We considered addressing this issue in our future research to explain the supply and demand relationship between user groups and urban green space in specific terms.
- The English editing and typos errors need to be improved.
Response 7: Thank you. We have further checked the full text and typography of this paper and asked a professional organization to polish the full text of the paper.

Round 2
Reviewer 3 Report
Several minor suggestions:
1. Please carefully check the typos problems. For example, in sections 2.3 and 3.1, the format of some subtitles is confusing: Line 269 "1. External spatial features". Meanwhile, "8/10 March 2018" in Table 1 would better be put as "8 and 10 March 2018" to be more precise.
2. I think Section 2.4 Research Framework should be presented before Section 2.3 Methodology to give the readers first a general picture of what the study will be about. And then it will be logical to go further to talk about concretely what are the indicators and treatment.
3. Figure 2 “Evaluation index system of UGS vitality” needs to be more precise to avoid confusion. It could be "influence index..." like in the title of Section 4.2.
Author Response
- Please carefully check the typos problems. For example, in sections 2.3 and 3.1, the format of some subtitles is confusing: Line 269 "1. External spatial features". Meanwhile, "8/10 March 2018" in Table 1 would better be put as "8 and 10 March 2018" to be more precise.
Response 1: Thank you for your careful suggestions. We have modified the unclear expression and date format you mentioned to make it easier to understand.
- I think Section 2.4 Research Framework should be presented before Section 2.3 Methodology to give the readers first a general picture of what the study will be about. And then it will be logical to go further to talk about concretely what are the indicators and treatment.
Response 2: Thanks for your suggestion. This modification really makes the logic of the paper clearer. We have adjusted the order of Section2.3 and Section2.4.
- Figure 2 “Evaluation index system of UGS vitality” needs to be more precise to avoid confusion. It could be "influence index..." like in the title of Section 4.2.
Response 3: Thank you for your suggestion. We have adjusted the relevant wording.
